# On Fairness of Task Arithmetic:
# The Role of Task Vectors

**Laura Gomezjurado Gonzalez**[*†]
Stanford University

**Hiroki Naganuma**[*]
Université de Montréal, Mila

**Kotaro Yoshida**[*]
Institute of Science Tokyo

**Takafumi Horie**
Kyoto University

**Yuji Naraki**
Independent Researcher

**Ryotaro Shimizu**
ZOZO Research, UCSD

## Abstract

Model editing techniques, particularly task arithmetic with task vectors, offer an efficient alternative to full fine-tuning by enabling direct parameter updates through simple arithmetic operations. While this approach promises substantial computational savings, its impact on fairness has remained largely unexplored—despite growing concern over biased outcomes in high-stakes applications such as hate speech detection. In this work, we present the first systematic study of fairness in task arithmetic, benchmarking it against full fine-tuning (FFT) and Low-Rank Adaptation (LoRA). We evaluate across multiple language models and datasets using standard group fairness metrics, including Demographic Parity and Equalized Odds. Our analysis shows that task vectors can be tuned to achieve competitive accuracy while reducing disparities, and that merging subgroup-specific task vectors provides a practical mechanism for steering fairness outcomes. We further provide a theoretical bound linking task-vector scaling to fairness metrics, offering insight into the observed trade-offs. Together, these findings establish task arithmetic not only as a cost-efficient editing method but also as a fairness-aware alternative to existing adaptation techniques, laying the groundwork for responsible deployment of large language models. Our code is available at `https://anonymous.4open.science/status/fairness_task_vector-4F2F`

## 1 Introduction

As large language models (LLMs) see broader application, efficient techniques for adapting them to specific tasks become increasingly crucial. While some models have been distilled [Sanh et al., 2019b, Jiao et al., 2020, Turc et al., 2020] or are relatively small [Abdin et al., 2024], task-specific fine-tuning often demands substantial computational resources, prompting the development of parameter-efficient fine-tuning (PEFT) methods [Houlsby et al., 2019, Hu et al., 2022, Ben Zaken et al., 2022, Dettmers et al., 2023].

One notable example is Low-Rank Adaptation (LoRA) [Hu et al., 2022], which updates a compact set of parameters while leaving most of the original weights untouched, thus reducing training costs. Despite the popularity of PEFT methods, they do not resolve every challenge: in high-stakes tasks with imbalanced data, LoRA and similar approaches can preserve or even amplify biases, raising concerns about fairness [Ding et al., 2024b, Sap et al., 2019].

An alternative strategy that has recently drawn attention is model editing with task vectors [Ilharco et al., 2023, Zhang et al., 2024, Yoshida et al., 2025]. A task vector is defined as the parameter

---

[*]Alphabetical order, these authors contributed equally to this work.
[†]**Correspondence:** lpgomez@stanford.edu

39th Conference on Neural Information Processing Systems (NeurIPS 2025) Workshop: .

difference between a base pre-trained model $\theta_{\text{base}}$ and a fine-tuned model $\theta_{\text{task}}$. By adding or subtracting this vector within the original weight space (so-called "task arithmetic"), a user can edit or remove the corresponding task-specific behavior without further gradient-based training [Ilharco et al., 2023]. Moreover, scaling the task vector grants fine-grained control over the strength of the transferred capability. This approach represents a promising direction, as it directly manipulates parameters while avoiding a costly re-optimization of the entire model.

In addition to these computational benefits, prior work has suggested that separating and analyzing task vectors may enhance interpretability [Cerrato et al., 2025]. By isolating the weight updates associated with particular subgroups (e.g., racial or gender demographics), one can potentially trace how the model adapts to each subgroup. This feature is appealing for investigating biases arising from unequal representation in training data, as it highlights which groups require larger shifts in weight space. Nevertheless, open questions persist regarding how well this model editing using task-vector preserves or exacerbates fairness. For instance, improving performance for one demographic might degrade outcomes for another, and it is not yet clear how to balance trade-offs with established fairness metrics such as Demographic Parity (DPD) or Equalized Odds (EOD).

To address this gap, we systematically examine how task-vector editing compares to both traditional full-parameter fine-tuning (FFT) and LoRA, and we further explore whether injecting task vectors into an FFT model offers additional control over fairness. Our experiments focus on hate-speech detection on Llama-7B [Touvron et al., 2023] and DistilBERT [Sanh et al., 2019a] , measured by subgroup-specific accuracy and widely used fairness metrics. Our contributions and findings are summarized as follows:

- A thorough comparison of four algorithms (FFT, LoRA, model editing using task-vector, and a hybrid approach injecting task vectors into FFT) in terms of their effects on fairness metrics and overall performance (Figure 1)].
- An analysis showing that task vectors can substantially improve fairness while preserving accuracy, provided that their scalar coefficients are appropriately tuned (Figure 2).
- Evidence that merging task vectors for underrepresented subgroups with existing models can adjust fairness outcomes without incurring a significant accuracy drop (Figures 3(a), 3(b) and 4(a)).
- We provide a theoretical upper bound (Appendix B) linking task-vector scaling to fairness metrics, offering an analytical explanation for the fairness–accuracy trade-offs observed empirically.

Through this analysis, we illustrate how task vectors can reduce risks from a fairness perspective while taking advantage of their flexibility and interpretability as a model editing approach. These findings provide a foundation for extending task-vector-based methods to promote fair and responsible operation of large language models.

## 2  Preliminaries

In this section, we first provide an overview of the fundamental concept of task vectors and the procedure known as task arithmetic, which applies these vectors to edit model behavior. We then introduce methods for merging multiple task vectors into a single model.

**Task arithmetic.**  A task vector is defined as the difference in model parameters between a fine-tuned model on a given task and the original base model. Formally, if $\theta_{\text{base}}$ are the pre-trained weights and $\theta_{\text{task}}$ are the weights after fine-tuning on a task, then the task vector is: $\Delta\theta = \theta_{\text{task}} - \theta_{\text{base}}$ [Ilharco et al., 2023].

This vector represents a direction in weight space such that moving the base model's weights by $\Delta\theta$ steers the model to perform well on task. In other words, adding $\Delta\theta$ to $\theta_{\text{base}}$ yields a model with improved performance on the target task, without any additional training. Once computed, task vectors can be manipulated through simple arithmetic operations to edit model behavior directly in weight space [Ilharco et al., 2023, Ortiz-Jimenez et al., 2024]. Key operations include:

**Addition:** Given two task vectors $\Delta\theta_A$ and $\Delta\theta_B$ (for tasks A and B), their sum can be applied to the base model ($\theta_{\text{base}} + \Delta\theta_A + \Delta\theta_B$) to produce a model that

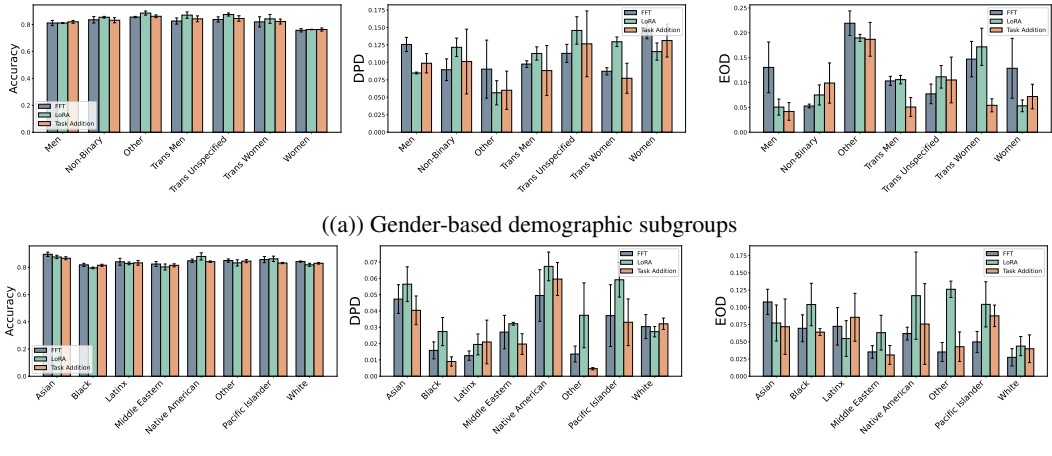

((a)) Gender-based demographic subgroups

((b)) Race-based demographic subgroups

Figure 1: LoRA and FFT vs. Task addition with the optimal coefficient for the training accuracy ($\lambda = 0.8$ for gender setting and $\lambda = 0.5$ for race setting) on group-wise accuracy, demographic parity difference (DPD, lower is fairer), and equalized odds difference (EOD, lower is fairer). Error bars denote the standard error across three seeds. Columns: group-wise accuracy, DPD, EOD. No consistent pattern emerges that task addition necessarily degrades subgroup fairness relative to LoRA or FFT subgroups show improvements or comparable results under task addition, while others show small declines.

exhibits improved performance on both tasks A and B [Ilharco et al., 2023]. This task addition effectively combines knowledge from multiple tasks into one model.

**Negation:** Using the negative of a task vector, $-\Delta\theta$, one can subtract a task's influence. For example, applying $\theta_{\text{base}} - \Delta\theta_A$ (or equivalently $\theta_{\text{base}} + (-\Delta\theta_A)$) yields a model with reduced performance on task A—effectively unlearning or forgetting it—while preserving other behaviors [Ilharco et al., 2023]. This is useful for removing undesirable skills or biases.

**Scalar scaling:** Multiplying a task vector by a scalar $\lambda$ adjusts the strength of the edit. For example, using $\theta_{\text{base}} + \lambda\Delta\theta_A$ allows partial ($0 < \lambda < 1$) or amplified ($\lambda > 1$) application of a task's effect. This scaling provides fine-grained control over how strongly the task knowledge is injected into the model.

**Merging task vectors.** Since task vectors reside in a common weight space, they can be merged by simple addition with tunable scaling. Formally, given a base model $\theta_0$ and task vectors $\Delta\theta_i$, one can construct a merged model as:

$$\theta_{\text{merged}} = \theta_0 + \sum_i \lambda_i \, \Delta\theta_i \, , \tag{1}$$

where each coefficient $\lambda_i$ controls the influence of task $i$. Varying $\lambda_i$ thus directly modulates how strongly the $i$-th task's knowledge is injected, allowing fine-grained blending of capabilities. Indeed, adding multiple task vectors with $\lambda_i = 1$ endows a model with all those capabilities simultaneously [Ilharco et al., 2023]. Optimizing the $\lambda_i$ values (i.e., learning an anisotropic scaling for each vector) further improves the composition by balancing contributions and reducing interference between tasks [Zhang et al., 2024].

## 3   Related Work

**Task arithmetic: efficiency and interpretability.** Task vectors offer a computationally efficient framework for editing and analyzing model behavior. Once a task vector is computed—namely, the weight difference between a base model and its fine-tuned variant [Ilharco et al., 2023, Zhang et al., 2024, Yoshida et al., 2025]—no additional training data or retraining is required to transfer or remove task-specific capabilities. By treating each fine-tuning update as a direction in weight

space, practitioners can combine or negate these updates through simple addition or subtraction [Ilharco et al., 2023]. This modularity not only reduces computational overhead but also enhances interpretability by isolating the contribution of each task.

Beyond modularity, task arithmetic can reveal valuable information about how and where a model adapts to new tasks. Li et al. [2024] show a near-linear relationship between data size and the norm of a task vector, suggesting that over-represented tasks can dominate weight space shifts in multi-task settings. In addition, the orientation of task vectors can indicate synergies or conflicts among tasks [Li et al., 2025], and decomposing these vectors by layer can pinpoint which parts of the model are most affected [Zhang et al., 2024, Gargiulo et al., 2025]. Hence, task vectors offer a promising lens for diagnosing training dynamics and identifying potential biases.

**Fairness metrics for LLMs.** Fairness in large language models is commonly evaluated using criteria such as Demographic Parity, Equalized Odds, and accuracy parity. Demographic Parity requires similar positive outcome rates across demographic groups, while Equalized Odds demands that true and false positive rates be equivalent. Accuracy parity checks for consistent predictive performance across groups [Fraenkel, 2020, Kennedy et al., 2020a, Pitoura, 2019, Quan et al., 2023]. These metrics are broadly used to detect biases and measure whether a model's behavior disproportionately disadvantages certain populations.

**FFT and LoRA under fairness constraints.** Parameter-efficient methods such as LoRA [Hu et al., 2022] address computational bottlenecks by training only a small set of parameters, yet they do not inherently solve fairness issues. In some cases, LoRA yields comparable subgroup performance to full fine-tuning [Ding et al., 2024b], while in others, it fails to mitigate toxic behaviors or biases [Das et al., 2024]. The variance in outcomes depends on factors like the rank of the LoRA matrices, the base model's quality, and the distribution of training data [Das et al., 2024].

**Merging tasks and fairness considerations.** Despite the potential efficiency gains and interpretability offered by task arithmetic, the merging of task vectors for multiple groups can trigger new challenges. For instance, simply summing vectors may lead to "negative transfer," where updates beneficial to one subgroup degrade performance for another [Ding et al., 2024a, Yu et al., 2020]. In highly imbalanced settings, merging models through supervised fine-tuning can also disproportionately favor majority groups while disadvantaging minorities [Cross et al., 2024]. Because fairness does not compose additively, interactions among subgroup-specific task vectors can produce unpredictable shifts in metrics like Demographic Parity and Equalized Odds [Gohar et al., 2023].

Consequently, identifying effective ways to adjust task vectors—such as through scalar scaling—remains a key step toward fairness-aware model editing. This work aims to fill that gap by systematically evaluating how these operations influence both fairness and overall model accuracy.

## 4 Experimental Setup

### 4.1 Configuration.

Building on the experimental framework established by Ding et al. [2024b], we adopted their evaluation and experimental procedure to assess the fairness implications of LoRA in comparison to FFT. In our work, we extend this analysis by focusing on how task arithmetic compares to both LoRA and FFT in terms of fairness and performance. The detailed experimental setup is provided in Appendix C.

**Datasets.** We use a modified version of the *Berkeley D-Lab Hate Speech* dataset originally introduced by Kennedy et al. [2020a] and adapted by Ding et al. [2024b], the research we are building upon. Our dataset contains a total of 6,898 tweet-sized text snippets annotated for hate speech and categorized by sensitive attributes: *Race* and *Gender*, each further divided into fine-grained subgroups (e.g., *Women*, *Non-binary*, *Men* within *Gender*) as shown in Table 1. We frame hate speech detection as a binary classification task: given a text snippet, the model predicts whether it constitutes hate speech (e.g., hatespeech in the Gender subset may target Non-binary or Trans Women). Each example includes both the hate speech label and one or more protected attribute annotations (e.g., *gender* = woman, *race* = Asian). These are used to assess subgroup-level performance and fairness metrics.

| Gender Subgroups | | Race Subgroups | |
|---|---|---|---|
| Men | 817 | Asian | 311 |
| Non-binary | 114 | Black | 1,007 |
| Trans men | 178 | Latinx | 368 |
| Trans unspecified | 173 | Native American | 153 |
| Trans women | 148 | Middle Eastern | 493 |
| Women | 2,057 | Pacific Islander | 138 |
| Other | 59 | White | 580 |
| | | Other | 302 |
| **Total** | **3,546** | **Total** | **3,352** |

Table 1: Data statistics in the gender and race subgroups.

This setting supports rigorous fairness analysis due to its rich attribute annotations and real-world relevance [Kennedy et al., 2020a]. To test generalization beyond hate speech, we apply our methods to the *Civil Comments* dataset [Borkan et al., 2019], a large-scale toxicity corpus with sensitive-attribute labels. We treat toxicity as binary with a $0.5$ threshold; comments above this are positive "flagged". Fairness is evaluated across Gender and Race subgroups.

**Evaluation metrics and fairness scope.**   Since we cast hate-speech and toxicity detection as binary classification, for each protected attribute (e.g., Gender, Race/Ethnicity), we compute subgroup-resolved metrics: *DPD* measures selection-rate disparity as the maximum absolute gap in flag rates across subgroups. *EOD* measures error-rate disparity by requiring both true-positive and false-positive rates to be comparable. *Accuracy-parity gap* is the maximum absolute difference in accuracy across subgroup pairs and serves as a stability indicator. We report per-subgroup values along with macro-averages and worst-group results. These choices mirror established practice and enable direct comparison to prior PEFT–fairness evaluations discussed in §**??**. Formal definitions and computation details appear in Appendix A.

## 4.2 Protocol.

We evaluate our methods using a main base model: LLaMA2-7B[3]. Our fairness evaluations focus on two sensitive attributes: gender and race, using subgroup-wise metrics mentioned earlier –accuracy, DPD, and EOD.

For FFT, the pretrained model was fine-tuned on the combined training data from all subgroups of the target attribute (gender or race). Evaluation was then performed on the test data from each corresponding subgroup, enabling fine-grained assessment of both performance and fairness.

For LoRA, we followed the same training and evaluation procedure as FFT. The rank of LoRA's adaptation modules was set to 8, following Ding et al. [2024b].

For task arithmetic, we applied a compositional fine-tuning approach. The training data was partitioned by subgroup (gender or race), and FFT was applied separately to each subgroup's data to produce fine-tuned models $\theta_i$. From these, we computed task vectors $\Delta\theta_i$ relative to the base model. These vectors were then merged using the approach described in Eq. (1), with a single, uniform scaling coefficient $\lambda$ applied to all vectors. $\lambda$ served as the sole hyperparameter in the merging process and was tuned on the training data. The evaluation metrics were computed in the same manner as for FFT and LoRA.

**Task vector coefficient adjustment.**   Building on the task vector merging framework introduced in Eq. (1), we further explore the impact of the scaling coefficient $\lambda$ on fairness outcomes. Specifically, we vary the uniform task vector coefficient $\lambda$ across a broad range (from 0.0 to 1.0 with 0.1 intervals) and evaluate how this adjustment influences subgroup-level fairness metrics, including accuracy, DPD, and EOD.

---

[3]LLaMA 2 is licensed under the LLAMA 2 Community License, Copyright (c) Meta Platforms, Inc. All Rights Reserved. See: https://ai.meta.com/llama/license

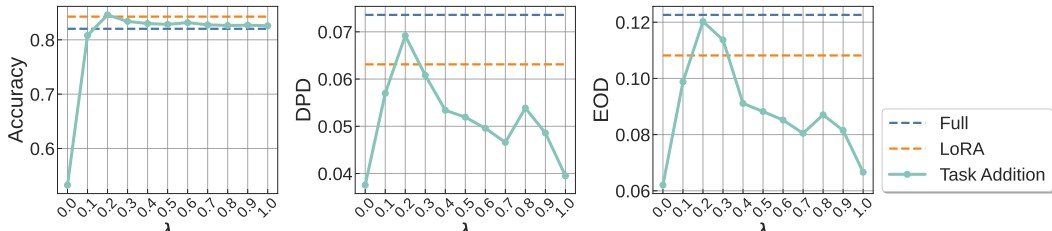

Figure 2: Varying the task arithmetic coefficient $\lambda$ and comparing against FFT (purple dashed) and LoRA (orange dashed) for macro-averaged accuracy (left), demographic parity difference (DPD, center), and equalized odds difference (EOD, right) on the **gender** subset. Higher accuracy is better; lower DPD/EOD indicate improved group fairness. For $\lambda \gtrsim 0.3$, task addition maintains *competitive accuracy* while *typically lowering* DPD/EOD relative to both baselines.

**Impact of worst-performing subgroup task vectors on fairness and performance.** To investigate whether incorporating task vectors from underperforming subgroups can improve fairness without sacrificing overall performance, we first identified the lowest-performing subgroups within each attribute based on the average of DPD and EOD under the FFT setting. We excluded the "others" group from this analysis as it does not reflect the characteristics of any specific subgroup. This selection was informed by both our experimental results and those reported in Ding et al. [2024b], which showed consistent patterns. For gender, the worst-performing subgroups were men and women; for race, they were Asian and Native American. We constructed a new model variant by injecting a worst-performing subgroup task vector worst-performing subgroup task vector into the base fine-tuned model:

$$\theta_{\text{new}} = \theta_{\text{SFT}} + \lambda(\theta_{\text{worst-performing subgroup}} - \theta_0)$$

where $\lambda$ controls the strength of the task vector injection. We varied $\lambda$ from from 0.0 to 1.0 at 0.2 intervals to analyze the effect of this targeted addition on subgroup fairness metrics and overall accuracy.

**Statistical Significance.** All results are averaged over three random seeds; we also compute 95% confidence intervals for key metrics to assess robustness (See Table 4).

## 5 Results

### 5.1 Theoretical intuition.

We complement our empirical findings with an analytical upper bound that links task-vector scaling to fairness metrics.

**Theorem (informal).** *Consider the merged model $\theta(\lambda) = \theta_0 + \sum_g \lambda \, \Delta\theta_g$, where $\Delta\theta_g$ denotes the task vector for subgroup g. Then the demographic parity difference (DPD) satisfies*

$$\text{DPD}(\theta(\lambda)) \ \leq \ 2L \sum_g \big|\lambda - 1\big| \, \|\Delta\theta_g\|_2, \quad \textit{for a Lipschitz constant L.}$$

Intuitively, deviations of the scaling coefficient $\lambda$ from the balanced setting ($\lambda = 1$) enlarge disparities in proportion to the norms of subgroup task vectors. This explains why fairness disparities shrink as $\lambda \to 1$, consistent with the empirical trends observed in Figure 2. A full derivation and tighter constants are provided in Appendix B.

### 5.2 Empirical results overview.

Figures 1(a) and 1(b) compare FFT, LoRA, and task addition across gender and race subgroups for hate speech detection on LLaMA-2. For task addition, we selected $\lambda = 0.8$ for gender, $\lambda = 0.5$ for race, as it achieved the highest average training accuracy across three random seeds within the tested range $\lambda \in [0.0, 1.0]$. These visualizations provide a direct comparison of subgroup-wise

model behavior. From the subgroup-level bar plots in Figure 1, we observe that accuracy remains consistently high and comparable across all three adaptation methods, regardless of subgroup. On *Civil Comments*, on both DistilBERT and Qwen-2.5, Task Addition reduces group disparities while keeping accuracy competitive. (see Appendix. E and Table 4 for full CIs/results).

We also observe that, relative to FFT, task addition improves fairness in five of seven gender subgroups and in three of eight race subgroups, with no single method dominating across all groups. The effect in fairness being subgroup-dependent, motivates treating $\lambda$ as a deliberate tuning knob and inspecting subgroup behavior explicitly. As shown in Appendix B.2, theoretically, task addition realizes a group-weighted ERM in the linearized model. Concretely, $\theta(\lambda) = \theta_0 + \sum_g \lambda_g \Delta\theta_g$ coincides with the one-step minimizer of a first-order surrogate where subgroup $g$ is re-weighted by $\lambda_g$. This explains the smooth fairness–utility frontier traced by sweeping $\lambda$, and Theorem 5.1 predicts larger parity swings for groups with larger $\|\Delta\theta_g\|_2$. The observed curves in Fig. 2 align with those predictions without further assumptions.

Taken together, the empirical trends and their first-order mechanism align with prior literature: our macro-averaged accuracy, DPD, and EOD findings for FFT and LoRA are consistent with [Ding et al., 2024b]. Moreover, the reductions perspective of Agarwal et al. [2018] and the equalized-odds criterion of ? anticipate precisely the trade-off behavior we document, reinforcing the robustness of our evaluation and interpretation.

### 5.3 Controlling accuracy and fairness metrics through lambda.

Figure 2 illustrates the overall performance of FFT, LoRA, and task arithmetic as the scaling coefficients for task addition vary from 0.0 to 1.0. We observe how varying the task-arithmetic coefficient $\lambda$ impacts macro-averaged accuracy (left), demographic parity difference (DPD, center), and equalized odds difference (EOD, right) on a gender subset of the data. As $\lambda$ increases from 0.0 to 0.2, we observe a peak in accuracy, but this configuration yields higher DPD and EOD, indicating reduced fairness. Beyond $\lambda = 0.3$, accuracy remains competitive compared to FFT and LoRA, while both DPD and EOD progressively decline, suggesting that fairness improves without severely compromising performance. Notably, these task addition curves stay consistently lower than FFT and LoRA in terms of DPD and EOD at higher $\lambda$ values. Overall, this ablation could indicate that tuning $\lambda$ provides a practical mechanism for balancing accuracy and fairness objectives, offering guidelines for practitioners who wish to fine-tune fairness outcomes while maintaining strong predictive performance.

### 5.4 Subgroup-targeted vectors: gains with trade-offs

To further analyze the effects of subgroup-specific task composition, Figure 3(a)–3(b) illustrate heatmaps where the y-axis lists each method or configuration under evaluation: FFT as baseline, followed by task arithmetic with varying scaling coefficients (0.0 to 1.0 with 0.2 intervals). The x-axis represents the subgroups— (e.g., Women, Trans, etc. for Gender). Each cell shows the corresponding performance metric (e.g., macro-averaged accuracy, DPD, or EOD for a given method on a specific subgroup. For these experiments, we added the task vector of the worst-performing subgroups (Women and Men for the gender dataset subset, and Asian, and Native American for the race dataset subset) to the FFT model, as explained earlier.

We generally observe that increasing the scaling coefficient $\lambda$ tends to improve overall accuracy, consistent with the trends observed in Figure 2. However, effects are not uniform across all subgroups. In the gender-based plots, for example, the Asian subgroup consistently achieves the highest accuracy and lowest DPD/EOD—highlighting a recurring tradeoff where performance gains for one group may exacerbate disparities for others. When the Women task vector is added (Figure 3(b)), accuracy improves for the Trans Women subgroups. However, fairness metrics for subgroups such as Men tend to worsen as the scaling coefficient $\lambda$ increases.

In Figure 3(a), injecting the Men task vector improves performance for some subgroups, yet Women consistently show lower accuracy and do not see consistent fairness improvements at higher $\lambda$. Some groups (e.g., Other, Trans Men, Trans Women) begin with relatively poor fairness under FFT and show partial improvements with task vector addition. Still, these improvements are not universal—for example, the Other subgroup often retains high EOD values regardless of $\lambda$. Likewise, Native American accuracy remains mostly unchanged across $\lambda$, while fairness metrics can deteriorate when

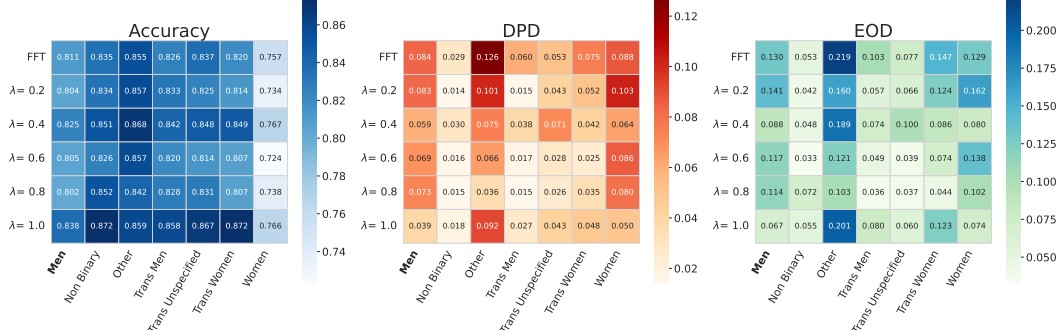

((a)) When **Men** task vector added to the FFT model on the **gender** subset.

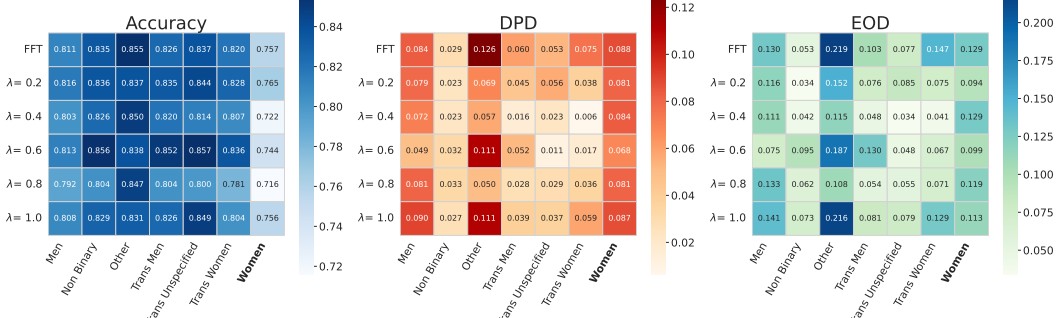

((b)) When **Women** task vector added to the FFT model on the **gender** subset.

Figure 3: Heatmaps of Accuracy (left), DPD (center), and EOD (right) for gender (top) and race (bottom) subgroups under the baseline FFT model ($\lambda = 0.0$) and with increasing $\lambda$ values from 0.2 to 1.0 in 0.2 increments. The task vector for Men was added on the gender subset (top), and the task vector for Women was added on the gender subset (bottom). Darker cells indicate higher values on each metric's scale; for DPD/EOD, lower values are better.

injecting task vectors for other groups. To visualize these results in more detail, Figure 4(a) shows macro-averaged accuracy, DPD, and EOD for the Men task vector added to the FFT model. The plots illustrate how varying the scaling coefficient $\lambda$ impacts overall performance and fairness, highlighting the effects of subgroup-specific task injection. We can observe in Figure 4(a) that injecting the Men task vector into the FFT model results in a slight accuracy gain and a clear monotonic decrease in both DPD and EOD as $\lambda$ increases—indicating a favorable and consistent improvement in fairness on the gender subset.

However, Figure 4(b) and the additional plots in Figures 10 and 11 in Appendix D.2 show more varied patterns as seen on Figures 3(a) and 3(b). When injecting the Native American task vector (Figure 11), accuracy remains stable while fairness seems to decrease (increased DPD and EOD). Asian (Figure 10) shows the same behavior as injecting the Men task vector (Figure 4(a)), positive increase of fairness metrics as $\lambda$ increases. These results show that injecting task vectors shifts fairness and performance in a group-specific manner, tracing a clear fairness–utility frontier. This heterogeneity is expected: per §5.2 and Theorem 5.1, sensitivity scales with $|\Delta\theta_g|_2$. Practically, task-vector merging thus offers a *subgroup-conditioned* control knob: identifying which $\Delta\theta_g$ help or hurt which groups provides a new actionable design consideration that SFT/LoRA do not expose, and that hasn't been explored in previous task arithmetic literature.

## 6 Conclusion and Limitations

**Conclusion.** In this study, we investigated the impact of a task arithmetic approach using task vectors on fairness, in comparison to conventional FFT and LoRA methods. We conducted detailed experiments to assess how the task addition affects prediction accuracy and fairness metrics, including the DPD and EOD across various subgroups. The results indicate that, with appropriate settings of

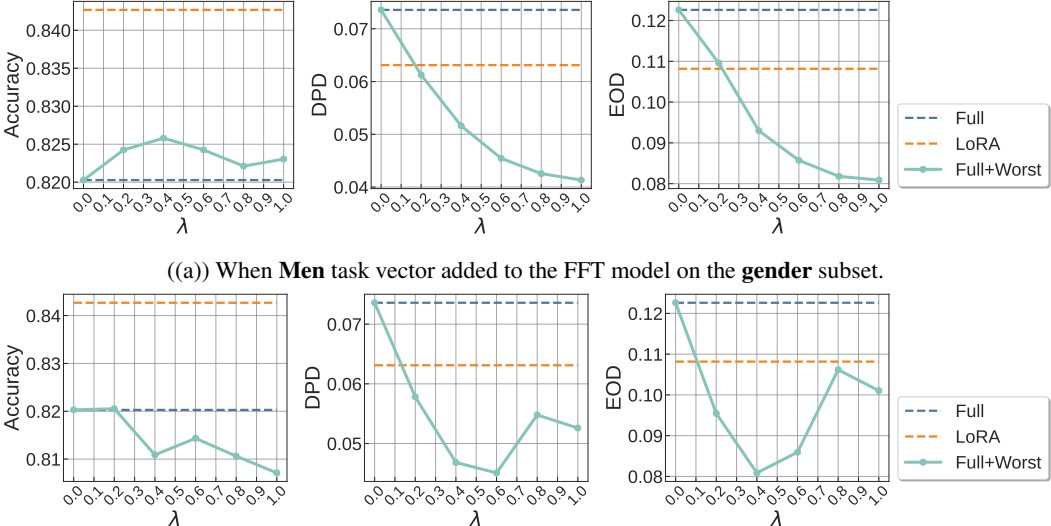

((a)) When **Men** task vector added to the FFT model on the **gender** subset.

((b)) When **Women** task vector added to the FFT model on the **gender** subset.

Figure 4: Impact of injecting both the **Men** and **Women** subgroup task vectors into the FFT model on the gender data subset. The plot illustrates how scaling coefficient $\lambda$ reduces DPD and EOD, outperforming the baseline FFT (blue dashed) and LoRA (orange dashed), with negligible impact on macro-averaged accuracy.

the scalar coefficient $\lambda$, the task arithmetic method can improve DPD and EOD without significantly compromising overall model accuracy. Notably, using low to moderate values of the task vector coefficient effectively reduced prediction bias in minority groups compared to FFT and LoRA.

Furthermore, the task arithmetic framework allows for subgroup-specific evaluation and adjustment of model updates, enhancing interpretability—a key advantage of this method in the context of fairness. This interpretability facilitates the mitigation of excessive bias or adverse effects on particular groups, ultimately enabling more balanced model training

Our theoretical bound on DPD B provides an interpretive lens: it explains why simple task addition can both reduce or worsen fairness depending on $\lambda$, since deviations from equal weighting directly increase disparities. This suggests future methods could optimize coefficients with fairness constraints in mind.

**Limitations.** Despite these promising results, several challenges remain. The effectiveness of task arithmetic depends on dataset characteristics and subgroup distributions, necessitating further investigation into its generalizability across different tasks and domains. Moreover, future work should explore algorithms for automatically optimizing the scalar coefficient $\lambda$ and for balancing trade-offs among multiple subgroups.

In summary, our study demonstrates that task arithmetic using task vectors offers a promising approach for controlling model fairness. Further experimental validation, application to diverse tasks, and developing trade-off optimization methods are essential for improving fairness in broader and more realistic deployment scenarios.

## Acknowledgement

The computation resource of this project is supported by SQUID cluster provided by Osaka University through the AI Research Projects. Hiroki Naganuma also acknowledges funding support from the ANRI Fellowship for this work.

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

# Appendix

# A    Fairness metrics

## A.1    Demographic Parity Difference (DPD) [Agarwal et al., 2018, 2019]

DPD measures how varied the model's rate of positive predictions is across attributes. This metric is calculated as follows:

$$M_{\text{DPD}} = \left| \Pr[f(X) = 1 \mid A = 1] - \Pr[f(X) = 1 \mid A = 0] \right|,$$

where $A$ is the sensitive attributes, $f(X)$ is the prediction from the models, and $X$ is the feature vector. The larger the DPD, the greater the difference in prediction outcomes across attributes, indicating greater unfairness in the model predictions.

## A.2    Equalized Odds Difference (EOD) [Ding et al., 2024b]

EOD is a metric that measures whether the model exhibits similar predictive performance in terms of true and false positives, regardless of the attribute.

$$M_{\text{eod}} = \max \left\{ M_{\text{TP}}, M_{\text{FP}} \right\}. \tag{2}$$

Here, letting $Y$ denote the true label, $M_{TP}$ and $M_{FP}$ are defined as follows:

$$M_{\text{TP}} = \left| \Pr[f(X) = 1 \mid Y = 1, A = 1] - \Pr[f(X) = 1 \mid Y = 1, A = 0] \right|,$$

$$M_{\text{FP}} = \left| \Pr[f(X) = 1 \mid Y = 0, A = 1] - \Pr[f(X) = 1 \mid Y = 0, A = 0] \right|.$$

## A.3    Accuracy Parity

Accuracy parity refers to the expectation that a classifier achieves comparable accuracy across different sensitive attribute groups. Formally, accuracy parity is satisfied when the probability of correct classification is equal across groups, i.e.,

$$\mathbb{E}(Y = \hat{Y} \mid S = 0) = \mathbb{E}(Y = \hat{Y} \mid S = 1), \tag{3}$$

This notion of fairness ensures that all subgroups receive equally reliable predictions, and is particularly relevant in applications where consistent model performance across demographics is critical. Unlike statistical parity or equal opportunity, accuracy parity focuses on equal overall correctness rather than specific error types or outcome rates [Quan et al., 2023].

We observed **high degree of accuracy parity** in both gender and race settings, as the accuracy differences between subgroups are negligible, indicating that the model performs consistently across all groups.

## B  DPD Upper Bound and Optimal Task–Vector Scaling

### B.1  Notation and Assumptions

**A1  Smooth predictions.** Soft scores $p_\theta$ satisfy $|p_\theta(x) - p_{\theta'}(x)| \le L \, \|\theta - \theta'\|_2 \; \forall x$.

**A2  Task vectors.** For each group $g \in \{1, \ldots, G\}$, $\Delta\theta_g := \theta_0^{(g)} - \theta_0$ is obtained with the *same* learning rate and schedule.

**A3  Scaling coefficients.** Coefficients obey $\sum_{g=1}^{G} \lambda_g = G$.

**A4  Symmetric data-generating process.** The joint distribution satisfies $\mathcal{D} = \bigcup_g \mathcal{D}_g$ where all $\mathcal{D}_g$ share the same conditional distribution except for the sensitive attribute label.

The merged model is

$$\theta(\boldsymbol{\lambda}) \; = \; \theta_0 + \sum_{g=1}^{G} \lambda_g \, \Delta\theta_g.$$

Demographic Parity Difference (DPD) reads

$$\mathrm{DPD}(\theta) \; = \; \left| \mathbb{E}_{\mathcal{D}_1}[p_\theta] - \mathbb{E}_{\mathcal{D}_0}[p_\theta] \right|.$$

### B.2  Task Addition and Weighted ERM

**Lemma 1** (First-order link). *Let $\ell(\theta; x)$ be the training loss. For any non-negative $\{\lambda_g\}$,*

$$\theta(\boldsymbol{\lambda}) \; \approx \; \arg\min_\theta \sum_{g=1}^{G} \lambda_g \, \mathbb{E}_{x \sim \mathcal{D}_g} \big[ \ell(\theta_0; x)$$
$$+ \, \nabla_\theta \ell(\theta_0; x)^\top (\theta - \theta_0) \big].$$

*That is, task addition gives the* first-order *solution of a group-weighted ERM.*

*Proof.* Insert the linear Taylor expansion of $\ell$ at $\theta_0$ and minimise the resulting quadratic form; the solution is exactly $\theta(\boldsymbol{\lambda})$. □

**Implication.** Deviation $|\lambda_g - 1|$ alters the group weights and therefore *directly pushes DPD upward*, as made explicit in Proposition 1 below.

### B.3  DPD Upper Bound

**Proposition 1** (DPD bound). *Under Assumptions **A1–A4**,*

$$\mathrm{DPD}\big(\theta(\boldsymbol{\lambda})\big) \; \le \; 2L \sum_{g=1}^{G} |\lambda_g - 1| \, \|\Delta\theta_g\|_2.$$

*Proof.* Define $\bar{\theta} := \theta_0 + \frac{1}{G}\sum_g \Delta\theta_g$. Assumption **A4** gives $\mathrm{DPD}(\bar{\theta}) = 0$. Put $f(x) := p_{\theta(\boldsymbol{\lambda})}(x) - p_{\bar{\theta}}(x)$. Then $\mathrm{DPD}(\theta(\boldsymbol{\lambda})) = |\mathbb{E}_{\mathcal{D}_1}[f] - \mathbb{E}_{\mathcal{D}_0}[f]|$. Triangle and Jensen yield $\leq 2L\,\|\theta(\boldsymbol{\lambda}) - \bar{\theta}\|_2$. Finally, $\theta(\boldsymbol{\lambda}) - \bar{\theta} = \sum_g (\lambda_g - 1)\Delta\theta_g$ and the triangle inequality give the stated bound. $\qquad\square$

# C   Experimental details

## C.1   Computational Resources and Software Environment

**Hardware and Software:** All experiments presented in this study were performed using computational resources equipped with two NVIDIA H100 GPUs. The experiments leveraged a GPU environment consisting of CUDA 12.1.0, cuDNN 9.0.0, and NCCL 2.20.5.

The experiments were conducted using Python 3.9.18, incorporating several essential Python libraries specifically optimized for deep learning tasks. The primary libraries included PyTorch (version 2.6.0), transformers (version 4.49.0), tokenizers (version 0.21.1), DeepSpeed (version 0.16.4), and Accelerate (version 1.5.2).

The training experiments utilized the DeepSpeed framework with the following key configurations: a gradient accumulation step of 4, optimizer offloaded to the CPU, zero redundancy optimizer at stage 2 (ZeRO-2), and mixed precision training employing FP16 and BF16 for enhanced performance and memory efficiency. All experiments were conducted with a total computational cost of approximately 30 GPU-hours.

**Protocol:** We fine-tuned models based on the Llama-7B [Touvron et al., 2023] architecture obtained via HuggingFace repositories. Each model was trained for 4 epochs, employing a cosine learning rate scheduler with a learning rate of $1 \times 10^{-5}$, a warm-up ratio of 0.01, and a weight decay of 0.001. Training utilized a per-device batch size of 2, with an effective batch size of 16 achieved through gradient accumulation. Reproducibility was ensured by setting a random seed of 13, 14, 15 across all experiments.

For Qwen2.5 experiments, models were trained for 2 epochs using a learning rate of $2 \times 10^{-5}$, a batch size of 16, and a sample fraction of 25% of the Civil Comments dataset. DistilBERT experiments utilized 2 epochs with a learning rate of $1 \times 10^{-5}$, a batch size of 16, and the full dataset (100% sample fraction). Both architectures employed a weight decay of 0.01 and evaluation/save strategies set to "epoch" with early stopping enabled.

For Low-Rank Adaptation (LoRA) experiments were conducted with a rank (lora_r) of 8, scaling factor (lora_alpha) of 16, and no dropout.

## C.2   Dataset

We use the Berkeley D-Lab hatespeech detection dataset [Kennedy et al., 2020b] [4] for our experiments.

The dataset is divided into subgroups based on the following attributes: *Race or Ethnicity*, *Religion*, *National Origin or Citizenship Status*, *Gender Identity*, *Sexual Orientation*, *Age*, and *Disability Status*. In our study, we use some of these subgroups to evaluate fairness.

Following Das et al. [2024], we binarize the hate speech score associated with each review using a threshold of 0.5 to determine whether the review constitutes hate speech. When multiple annotations exist for the same instance, we obtain one human annotation to avoid duplication.

# D   Additional Results

Here, we present results focusing on diverse subgroups, which we could not include in the main paper due to space constraints.

## D.1   Comparison of FFT, LoRA, and Task Arithmetic

Figure 7 illustrates the overall performance of FFT, LoRA, and task arithmetic as the scaling for task arithmetic vary from 0.0 to 1.0. Trends observed reinforced results on the gender subset on

---

[4] https://huggingface.co/datasets/ucberkeley-dlab/measuring-hate-speech

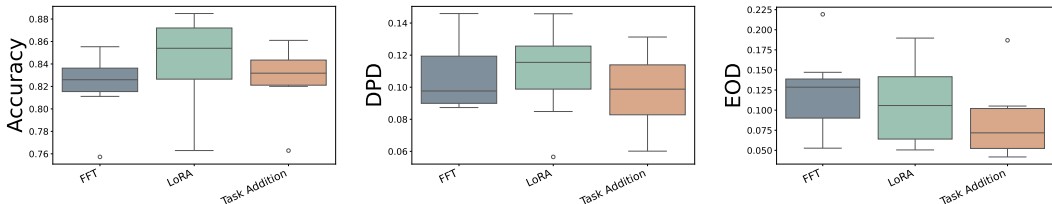

Figure 5: Boxplots of group-wise accuracy, demographic parity difference (DPD), and equalized odds difference (EOD) for —FFT, LoRA, and task addition with coefficient ($\lambda = 0.8$) —evaluated on the **gender** subset of the data. Higher accuracy is desirable, whereas lower DPD and EOD values indicate improved fairness. Boxplots show medians, interquartile ranges, and variability (with standard error across three seeds). While accuracy is similar across methods, Task Addition generally yields lower DPD and EOD medians than FFT and LoRA, suggesting a better balance between performance and fairness, though overlapping distributions imply these differences are not uniformly significant.

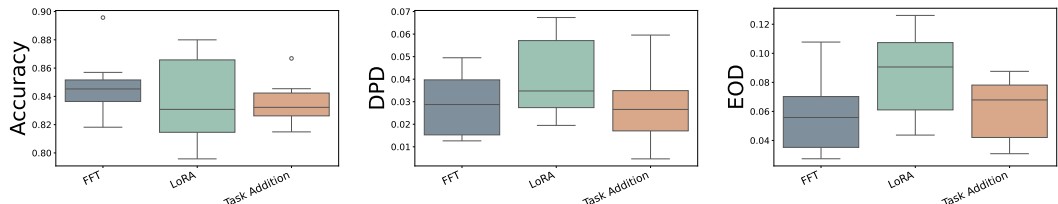

Figure 6: Boxplots of group-wise accuracy, demographic parity difference (DPD), and equalized odds difference (EOD) for —FFT, LoRA, and Task Addition with optimal coefficient ($\lambda = 0.5$) —evaluated on the **race** subset of the data. Higher accuracy is desirable, whereas lower DPD and EOD values indicate improved fairness. Boxplots show medians, interquartile ranges, and variability (with standard error across three seeds).

Figure 2. Overall, $\lambda$ provides a practical mechanism for balancing accuracy and fairness objectives, and similarly there is a peak at $\lambda = 0.2$ for highest accuracy, and higher DPD and EOD (less fairness).

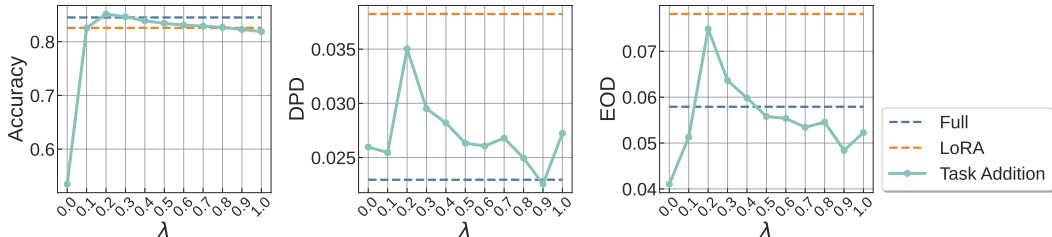

Figure 7: On a **race-focused** subset, we vary task arithmetic's coefficient $\lambda$ and compare it against FFT (purple dashed) and LoRA (orange dashed). The plots show group-wise accuracy (left), demographic parity difference (DPD, center), and equalized odds difference (EOD, right). Higher accuracy is better, while lower DPD and EOD indicate improved fairness. As $\lambda$ changes, task arithmetic remains competitive in accuracy and can reduce fairness gaps relative to the baselines.

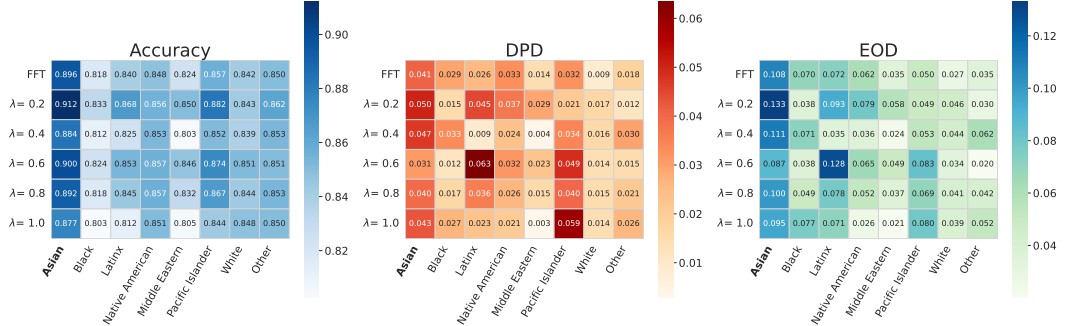

Figure 8: The task vector corresponding to **Asian** was added to the FFT model on the race data subset. Heatmap of Accuracy (left), DPD (center), and EOD (right) under the baseline (FFT) and increasing λ values (0.2 to 1.0). Darker cells indicate higher values in each metric's scale; for DPD/EOD, lower is better.

## D.2 Subgroup-Specific Task Addition to FFT

We include additional heatmaps that visualize subgroup-wise performance across FFT and varying scaling coefficients for the FFT model injected with a worst-performing subgroup. These supplementary plots, which follow the same setup described earlier, are consistent with the trends observed in Figures 3(a)–3(b).

In both gender and race subgroup experiments, increasing the scaling coefficient λ generally leads to improved macro-averaged accuracy. However, its impact on fairness metrics—DPD and EOD—is less predictable and varies across subgroups. For instance, some subgroups benefit from improved fairness as their corresponding task vectors are added, while others experience increased disparity, even if accuracy remains stable or improves.

This nuanced behavior reflects a broader pattern: gains in performance for certain subgroups can sometimes come at the expense of fairness for others. Injecting task vectors from worst-performing subgroups does not consistently reduce disparities and, in some cases, can amplify them.

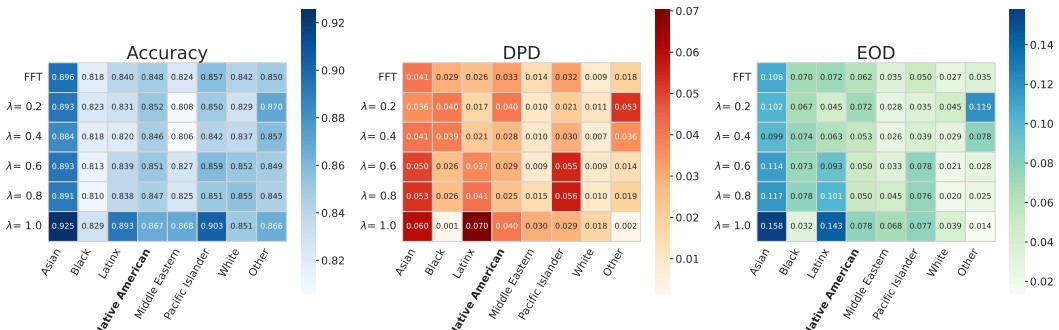

Figure 9: The task vector corresponding to **Native American** was added to the FFT model on the race data subset. Heatmap of Accuracy (left), DPD (center), and EOD (right) under the baseline (FFT) and increasing λ values (0.2 to 1.0). Darker cells indicate higher values in each metric's scale; for DPD/EOD, lower is better.

Figures 11–4(b) present additional results for the Full+Worst configuration, in which task vectors from the worst-performing subgroups (Native American, Asian, Men, and Women) are added to the FFT model. These plots show macro-averaged accuracy, DPD, and EOD as a function of the scaling coefficient λ.

Across these figures, we observe mixed effects: while accuracy generally remains stable or improves slightly, fairness outcomes vary by subgroup. In Figure 11, DPD and EOD worsen despite minimal accuracy changes. Meanwhile, Figure 4(b) reveals stable performance with minor fairness improvements, though gains are not consistent across metrics. These results further emphasize that

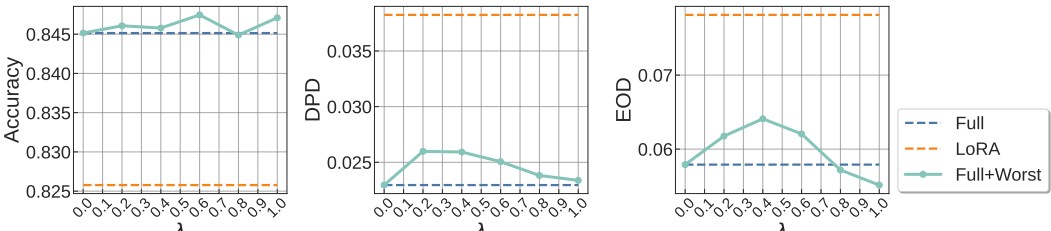

Figure 10: Effect of adding the **Asian** task vector to the FFT model on the **race** subset. Accuracy keeps competitive with increasing $\lambda$, and both DPD and EOD decrease consistently.

task vector injection alone does not ensure universal fairness improvements and often introduces subgroup-specific trade-offs.

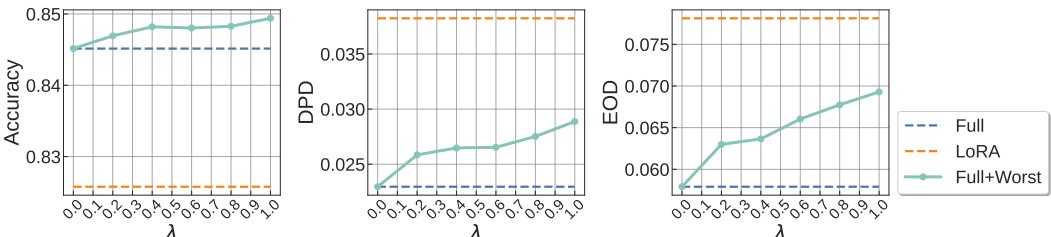

Figure 11: Results of injecting the **Native American** task vector into the FFT model. Accuracy shows minimal change across $\lambda$, while DPD and EOD increase (worsen fairness).

# E  Additional Experiments on Civil Comments

**Protocol & uncertainty.**   Unless noted, we follow the LLaMA-2 setup (Section 4.2): SFT and LoRA ($r=8$) to obtain subgroup-specific models, compute task vectors w.r.t. the pretrained base, and merge with a uniform scalar $\lambda$. We sweep $\lambda$ on the validation split (maximize overall accuracy) and evaluate on the test split. Uncertainty is 95% stratified bootstrap over the test set (2,000 resamples, preserving group $\times$ label frequencies). When multiple seeds are used, we pool predictions before resampling. For accuracy, we additionally report Wilson CIs when relevant.

**At a glance.**   On Civil Comments with DistilBERT (67M), task addition maintains accuracy within ∼0.6–1.1pp of SFT/LoRA while reducing fairness gaps: for *gender*, DPD drops by ≈41–54% and EOD by ≈34–47%; for *race*, DPD drops by ≈41–58% and EOD by ≈58–73% (midpoint comparisons). These patterns align with LLaMA-2 on the Berkeley D-Lab dataset (Table 4). As a complementary cross-architecture check, Qwen-2.5-0.5B on *gender* exhibits the same qualitative $\lambda$-controlled trade-off, improving substantially over LoRA with competitive accuracy.

## E.1  Civil Comments — Gender

**Notes.**   Relative to LoRA, Qwen-2.5-0.5B task addition halves DPD/EOD (∼54–56%) while regaining ∼3.3pp accuracy; relative to SFT, accuracy is lower and fairness is mixed (DPD comparable; EOD higher). DistilBERT shows consistent reductions in DPD/EOD with ≲1pp accuracy cost.

## E.2  Civil Comments — Race

**Discussion.**   Together with LLaMA-2 on Berkeley D-Lab (Table 4), these experiments indicate that the $\lambda$-controlled fairness–utility trade-off extends across architectures and datasets: task addition typically preserves accuracy within ∼1pp while materially reducing worst-case DPD/EOD.

  1. **Claims**

Table 2: **Civil Comments (Gender).** Headline metrics (Accuracy ↑, worst-case DPD ↓, worst-case EOD ↓). Entries are 95% CIs from stratified bootstrap; point estimates marked with † will be replaced by CIs computed using the same protocol.

| Model/Method | Accuracy | Worst-DPD | Worst-EOD |
|---|---|---|---|
| DistilBERT  SFT | 0.9457–0.9476 | 0.0887–0.1101 | 0.6157–0.6433 |
| DistilBERT  LoRA | 0.9447–0.9453 | 0.0735–0.0812 | 0.5024–0.5084 |
| **DistilBERT  Task Addition** | **0.9395**† | **0.0454**† | **0.3358**† |
| Qwen-2.5-0.5B  SFT[1] | 0.884–0.886 | 0.093–0.119 | 0.060–0.084 |
| Qwen-2.5-0.5B  LoRA[1] | 0.774–0.790 | 0.210–0.251 | 0.232–0.362 |
| **Qwen-2.5-0.5B  Task Addition**[1] | **0.810–0.820** | **0.100–0.103** | **0.130–0.143** |

† Point estimates; CIs to be computed with the same bootstrap.

Table 3: **Civil Comments (Race).** Headline metrics (Accuracy ↑, worst-case DPD ↓, worst-case EOD ↓). Models evaluated for this attribute are shown. CIs are 95% stratified bootstrap; † indicates point estimates to be replaced by CIs.

| Model/Method | Accuracy | Worst-DPD | Worst-EOD |
|---|---|---|---|
| DistilBERT  SFT | 0.9467–0.9473 | 0.0987–0.0995 | 0.2568–0.3544 |
| DistilBERT  LoRA | 0.9446–0.9453 | 0.1360–0.1425 | 0.4649–0.4895 |
| **DistilBERT  Task Addition** | **0.9362**† | **0.0580**† | **0.1289**† |

Question: Do the main claims made in the abstract and introduction accurately reflect the paper's contributions and scope?

Answer: [Yes]

Justification: The abstract and introduction clearly state the contributions: systematic evaluation of fairness in task arithmetic, comparison to FFT and LoRA, and analysis of subgroup-specific task vectors. The results sections directly support these claims.

Guidelines:

- The answer NA means that the abstract and introduction do not include the claims made in the paper.
- The abstract and/or introduction should clearly state the claims made, including the contributions made in the paper and important assumptions and limitations. A No or NA answer to this question will not be perceived well by the reviewers.
- The claims made should match theoretical and experimental results, and reflect how much the results can be expected to generalize to other settings.
- It is fine to include aspirational goals as motivation as long as it is clear that these goals are not attained by the paper.

2. **Limitations**

Question: Does the paper discuss the limitations of the work performed by the authors?

Answer: [Yes]

Justification: A dedicated "Limitations" section is included. It discusses dependence on dataset characteristics, subgroup distributions, and the need for future methods to optimize coefficients automatically.

Guidelines:

- The answer NA means that the paper has no limitation while the answer No means that the paper has limitations, but those are not discussed in the paper.
- The authors are encouraged to create a separate "Limitations" section in their paper.
- The paper should point out any strong assumptions and how robust the results are to violations of these assumptions (e.g., independence assumptions, noiseless settings, model well-specification, asymptotic approximations only holding locally). The authors should reflect on how these assumptions might be violated in practice and what the implications would be.

| Model | Race (95% CI) | Accuracy | Worst DPD | Worst EOD |
|---|---|---|---|---|
| LLaMA2-7B | SFT | 0.7901–0.9039 | 0.0000–0.0345 | 0.0000–0.0730 |
| | LoRA | 0.7599–0.9143 | 0.0000–0.0459 | 0.0000–0.1087 |
| | Task addition | **0.7972–0.8724** | **0.0000–0.0265** | **0.0000–0.1308** |
| DistilBERT | SFT | 0.9467–0.9473 | 0.0987–0.0995 | 0.2568–0.3544 |
| | LoRA | 0.9446–0.9453 | 0.1360–0.1425 | 0.4649–0.4895 |
| | Task addition | **0.9362** | **0.0580** | **0.1289** |
| **Model** | **Gender (95% CI)** | Accuracy | Worst DPD | Worst EOD |
| LLaMA2-7B | SFT | 0.7914–0.8491 | 0.0621–0.1125 | 0.0000–0.1794 |
| | LoRA | 0.8031–0.8823 | 0.0535–0.0596 | 0.0105–0.0906 |
| | Task addition | **0.8031–0.8823** | **0.0259–0.0943** | **0.0000–0.0858** |
| DistilBERT | SFT | 0.9457–0.9476 | 0.0887–0.1101 | 0.6157–0.6433 |
| | LoRA | 0.9447–0.9453 | 0.0735–0.0812 | 0.5024–0.5084 |
| | Task addition | **0.9395** | **0.0454** | **0.3358** |
| Qwen-2.5-0.5B[1] | SFT | 0.884–0.886 | 0.093–0.119 | 0.060–0.084 |
| Qwen-2.5-0.5B | LoRA | 0.774–0.790 | 0.210–0.251 | 0.232–0.362 |
| **Qwen-2.5-0.5B** | **Task addition** | **0.810–0.820** | **0.100–0.103** | **0.130–0.143** |

Table 4: 95% confidence intervals. Models evaluated for each attribute are shown: LLaMA2-7B on Berkeley D-Lab; DistilBERT and Qwen-2.5-0.5B on Civil Comments (Qwen-2.5 for gender). Task addition maintains accuracy while showing competitive or improved fairness compared to SFT and LoRA.

- The authors should reflect on the scope of the claims made, e.g., if the approach was only tested on a few datasets or with a few runs. In general, empirical results often depend on implicit assumptions, which should be articulated.
- The authors should reflect on the factors that influence the performance of the approach. For example, a facial recognition algorithm may perform poorly when image resolution is low or images are taken in low lighting. Or a speech-to-text system might not be used reliably to provide closed captions for online lectures because it fails to handle technical jargon.
- The authors should discuss the computational efficiency of the proposed algorithms and how they scale with dataset size.
- If applicable, the authors should discuss possible limitations of their approach to address problems of privacy and fairness.
- While the authors might fear that complete honesty about limitations might be used by reviewers as grounds for rejection, a worse outcome might be that reviewers discover limitations that aren't acknowledged in the paper. The authors should use their best judgment and recognize that individual actions in favor of transparency play an important role in developing norms that preserve the integrity of the community. Reviewers will be specifically instructed to not penalize honesty concerning limitations.

3. **Theory assumptions and proofs**

   Question: For each theoretical result, does the paper provide the full set of assumptions and a complete (and correct) proof?

   Answer: [Yes]

   Justification: The paper includes theoretical results in Appendix A (DPD upper bound, Lemma, Proposition) with explicit assumptions and complete proofs.

   Guidelines:

   - The answer NA means that the paper does not include theoretical results.
   - All the theorems, formulas, and proofs in the paper should be numbered and cross-referenced.
   - All assumptions should be clearly stated or referenced in the statement of any theorems.

- The proofs can either appear in the main paper or the supplemental material, but if they appear in the supplemental material, the authors are encouraged to provide a short proof sketch to provide intuition.
- Inversely, any informal proof provided in the core of the paper should be complemented by formal proofs provided in appendix or supplemental material.
- Theorems and Lemmas that the proof relies upon should be properly referenced.

4. **Experimental result reproducibility**

Question: Does the paper fully disclose all the information needed to reproduce the main experimental results of the paper to the extent that it affects the main claims and/or conclusions of the paper (regardless of whether the code and data are provided or not)?

Answer: [Yes]

Justification: Experimental setup is described in detail (Section 4, Appendix F): dataset, preprocessing, model architecture, hyperparameters, training protocol, evaluation metrics, and seeds are all disclosed.

Guidelines:

- The answer NA means that the paper does not include experiments.
- If the paper includes experiments, a No answer to this question will not be perceived well by the reviewers: Making the paper reproducible is important, regardless of whether the code and data are provided or not.
- If the contribution is a dataset and/or model, the authors should describe the steps taken to make their results reproducible or verifiable.
- Depending on the contribution, reproducibility can be accomplished in various ways. For example, if the contribution is a novel architecture, describing the architecture fully might suffice, or if the contribution is a specific model and empirical evaluation, it may be necessary to either make it possible for others to replicate the model with the same dataset, or provide access to the model. In general. releasing code and data is often one good way to accomplish this, but reproducibility can also be provided via detailed instructions for how to replicate the results, access to a hosted model (e.g., in the case of a large language model), releasing of a model checkpoint, or other means that are appropriate to the research performed.
- While NeurIPS does not require releasing code, the conference does require all submissions to provide some reasonable avenue for reproducibility, which may depend on the nature of the contribution. For example
  (a) If the contribution is primarily a new algorithm, the paper should make it clear how to reproduce that algorithm.
  (b) If the contribution is primarily a new model architecture, the paper should describe the architecture clearly and fully.
  (c) If the contribution is a new model (e.g., a large language model), then there should either be a way to access this model for reproducing the results or a way to reproduce the model (e.g., with an open-source dataset or instructions for how to construct the dataset).
  (d) We recognize that reproducibility may be tricky in some cases, in which case authors are welcome to describe the particular way they provide for reproducibility. In the case of closed-source models, it may be that access to the model is limited in some way (e.g., to registered users), but it should be possible for other researchers to have some path to reproducing or verifying the results.

5. **Open access to data and code**

Question: Does the paper provide open access to the data and code, with sufficient instructions to faithfully reproduce the main experimental results, as described in supplemental material?

Answer: [Yes]

Justification: The datasets used (Berkeley D-Lab Hate Speech and Civil Comments) are publicly available on HuggingFace. We plan to release the Github link in the camera-ready version.

Guidelines:

- The answer NA means that paper does not include experiments requiring code.
- Please see the NeurIPS code and data submission guidelines (`https://nips.cc/public/guides/CodeSubmissionPolicy`) for more details.
- While we encourage the release of code and data, we understand that this might not be possible, so "No" is an acceptable answer. Papers cannot be rejected simply for not including code, unless this is central to the contribution (e.g., for a new open-source benchmark).
- The instructions should contain the exact command and environment needed to run to reproduce the results. See the NeurIPS code and data submission guidelines (`https://nips.cc/public/guides/CodeSubmissionPolicy`) for more details.
- The authors should provide instructions on data access and preparation, including how to access the raw data, preprocessed data, intermediate data, and generated data, etc.
- The authors should provide scripts to reproduce all experimental results for the new proposed method and baselines. If only a subset of experiments are reproducible, they should state which ones are omitted from the script and why.
- At submission time, to preserve anonymity, the authors should release anonymized versions (if applicable).
- Providing as much information as possible in supplemental material (appended to the paper) is recommended, but including URLs to data and code is permitted.

6. **Experimental setting/details**

Question: Does the paper specify all the training and test details (e.g., data splits, hyperparameters, how they were chosen, type of optimizer, etc.) necessary to understand the results?

Answer: [Yes]

Justification: Section 4 and Appendix F specify training/test splits, optimizer, learning rate, batch sizes, epochs, LoRA rank, and seeds, sufficient for understanding and reproducing results.

Guidelines:

- The answer NA means that the paper does not include experiments.
- The experimental setting should be presented in the core of the paper to a level of detail that is necessary to appreciate the results and make sense of them.
- The full details can be provided either with the code, in appendix, or as supplemental material.

7. **Experiment statistical significance**

Question: Does the paper report error bars suitably and correctly defined or other appropriate information about the statistical significance of the experiments?

Answer: [Yes]

Justification: Error bars (standard error across three seeds) are reported in figures, and the text explains variability across runs. This supports statistical robustness of results.

Guidelines:

- The answer NA means that the paper does not include experiments.
- The authors should answer "Yes" if the results are accompanied by error bars, confidence intervals, or statistical significance tests, at least for the experiments that support the main claims of the paper.
- The factors of variability that the error bars are capturing should be clearly stated (for example, train/test split, initialization, random drawing of some parameter, or overall run with given experimental conditions).
- The method for calculating the error bars should be explained (closed form formula, call to a library function, bootstrap, etc.)
- The assumptions made should be given (e.g., Normally distributed errors).
- It should be clear whether the error bar is the standard deviation or the standard error of the mean.

- It is OK to report 1-sigma error bars, but one should state it. The authors should preferably report a 2-sigma error bar than state that they have a 96% CI, if the hypothesis of Normality of errors is not verified.
- For asymmetric distributions, the authors should be careful not to show in tables or figures symmetric error bars that would yield results that are out of range (e.g. negative error rates).
- If error bars are reported in tables or plots, The authors should explain in the text how they were calculated and reference the corresponding figures or tables in the text.

8. **Experiments compute resources**

   Question: For each experiment, does the paper provide sufficient information on the computer resources (type of compute workers, memory, time of execution) needed to reproduce the experiments?

   Answer: [Yes]

   Justification: Appendix F describes compute environment: $2 \times$ NVIDIA H100 GPUs, CUDA/cuDNN versions, 30 GPU-hours total, and resource details for training runs.

   Guidelines:
   - The answer NA means that the paper does not include experiments.
   - The paper should indicate the type of compute workers CPU or GPU, internal cluster, or cloud provider, including relevant memory and storage.
   - The paper should provide the amount of compute required for each of the individual experimental runs as well as estimate the total compute.
   - The paper should disclose whether the full research project required more compute than the experiments reported in the paper (e.g., preliminary or failed experiments that didn't make it into the paper).

9. **Code of ethics**

   Question: Does the research conducted in the paper conform, in every respect, with the NeurIPS Code of Ethics https://neurips.cc/public/EthicsGuidelines?

   Answer: [Yes]

   Justification:The study conforms to the NeurIPS Code of Ethics: it uses open, licensed datasets, no private or personally identifiable information, and explicitly investigates fairness and bias mitigation.

   Guidelines:
   - The answer NA means that the authors have not reviewed the NeurIPS Code of Ethics.
   - If the authors answer No, they should explain the special circumstances that require a deviation from the Code of Ethics.
   - The authors should make sure to preserve anonymity (e.g., if there is a special consideration due to laws or regulations in their jurisdiction).

10. **Broader impacts**

    Question: Does the paper discuss both potential positive societal impacts and negative societal impacts of the work performed?

    Answer: [Yes]

    Justification: The paper discusses both positive impacts (fairness-aware model editing, interpretability) and negative risks (potential misuse or subgroup tradeoffs) in the Conclusion and Limitations sections.

    Guidelines:
    - The answer NA means that there is no societal impact of the work performed.
    - If the authors answer NA or No, they should explain why their work has no societal impact or why the paper does not address societal impact.
    - Examples of negative societal impacts include potential malicious or unintended uses (e.g., disinformation, generating fake profiles, surveillance), fairness considerations (e.g., deployment of technologies that could make decisions that unfairly impact specific groups), privacy considerations, and security considerations.

- The conference expects that many papers will be foundational research and not tied to particular applications, let alone deployments. However, if there is a direct path to any negative applications, the authors should point it out. For example, it is legitimate to point out that an improvement in the quality of generative models could be used to generate deepfakes for disinformation. On the other hand, it is not needed to point out that a generic algorithm for optimizing neural networks could enable people to train models that generate Deepfakes faster.
- The authors should consider possible harms that could arise when the technology is being used as intended and functioning correctly, harms that could arise when the technology is being used as intended but gives incorrect results, and harms following from (intentional or unintentional) misuse of the technology.
- If there are negative societal impacts, the authors could also discuss possible mitigation strategies (e.g., gated release of models, providing defenses in addition to attacks, mechanisms for monitoring misuse, mechanisms to monitor how a system learns from feedback over time, improving the efficiency and accessibility of ML).

11. **Safeguards**

Question: Does the paper describe safeguards that have been put in place for responsible release of data or models that have a high risk for misuse (e.g., pretrained language models, image generators, or scraped datasets)?

Answer: [NA]

Justification: The paper does not release new pretrained models or datasets at high risk of misuse; it evaluates fairness on an existing public dataset.

Guidelines:

- The answer NA means that the paper poses no such risks.
- Released models that have a high risk for misuse or dual-use should be released with necessary safeguards to allow for controlled use of the model, for example by requiring that users adhere to usage guidelines or restrictions to access the model or implementing safety filters.
- Datasets that have been scraped from the Internet could pose safety risks. The authors should describe how they avoided releasing unsafe images.
- We recognize that providing effective safeguards is challenging, and many papers do not require this, but we encourage authors to take this into account and make a best faith effort.

12. **Licenses for existing assets**

Question: Are the creators or original owners of assets (e.g., code, data, models), used in the paper, properly credited and are the license and terms of use explicitly mentioned and properly respected?

Answer: [Yes]

Justification: The Berkeley D-Lab dataset is cited and linked (HuggingFace), with licensing acknowledged. LLaMA2 is used under its community license, which is explicitly referenced.

Guidelines:

- The answer NA means that the paper does not use existing assets.
- The authors should cite the original paper that produced the code package or dataset.
- The authors should state which version of the asset is used and, if possible, include a URL.
- The name of the license (e.g., CC-BY 4.0) should be included for each asset.
- For scraped data from a particular source (e.g., website), the copyright and terms of service of that source should be provided.
- If assets are released, the license, copyright information, and terms of use in the package should be provided. For popular datasets, `paperswithcode.com/datasets` has curated licenses for some datasets. Their licensing guide can help determine the license of a dataset.

- For existing datasets that are re-packaged, both the original license and the license of the derived asset (if it has changed) should be provided.
- If this information is not available online, the authors are encouraged to reach out to the asset's creators.

13. **New assets**

Question: Are new assets introduced in the paper well documented and is the documentation provided alongside the assets?

Answer: [NA]

Justification: The paper does not introduce new datasets, codebases, or pretrained models as assets.

Guidelines:

- The answer NA means that the paper does not release new assets.
- Researchers should communicate the details of the dataset/code/model as part of their submissions via structured templates. This includes details about training, license, limitations, etc.
- The paper should discuss whether and how consent was obtained from people whose asset is used.
- At submission time, remember to anonymize your assets (if applicable). You can either create an anonymized URL or include an anonymized zip file.

14. **Crowdsourcing and research with human subjects**

Question: For crowdsourcing experiments and research with human subjects, does the paper include the full text of instructions given to participants and screenshots, if applicable, as well as details about compensation (if any)?

Answer: [NA]

Justification: he work does not involve crowdsourcing or direct human-subject studies.

Guidelines:

- The answer NA means that the paper does not involve crowdsourcing nor research with human subjects.
- Including this information in the supplemental material is fine, but if the main contribution of the paper involves human subjects, then as much detail as possible should be included in the main paper.
- According to the NeurIPS Code of Ethics, workers involved in data collection, curation, or other labor should be paid at least the minimum wage in the country of the data collector.

15. **Institutional review board (IRB) approvals or equivalent for research with human subjects**

Question: Does the paper describe potential risks incurred by study participants, whether such risks were disclosed to the subjects, and whether Institutional Review Board (IRB) approvals (or an equivalent approval/review based on the requirements of your country or institution) were obtained?

Answer: [NA]

Justification: No human subjects were involved; the study used an existing public dataset.

Guidelines:

- The answer NA means that the paper does not involve crowdsourcing nor research with human subjects.
- Depending on the country in which research is conducted, IRB approval (or equivalent) may be required for any human subjects research. If you obtained IRB approval, you should clearly state this in the paper.
- We recognize that the procedures for this may vary significantly between institutions and locations, and we expect authors to adhere to the NeurIPS Code of Ethics and the guidelines for their institution.

- For initial submissions, do not include any information that would break anonymity (if applicable), such as the institution conducting the review.

16. **Declaration of LLM usage**

Question: Does the paper describe the usage of LLMs if it is an important, original, or non-standard component of the core methods in this research? Note that if the LLM is used only for writing, editing, or formatting purposes and does not impact the core methodology, scientific rigorousness, or originality of the research, declaration is not required.

Answer: [NA]

Justification: The paper uses large language models (LLaMA2-7B) as the core experimental subject, and their role is fully described in Section 4 and Appendix F.

Guidelines:

- The answer NA means that the core method development in this research does not involve LLMs as any important, original, or non-standard components.
- Please refer to our LLM policy (`https://neurips.cc/Conferences/2025/LLM`) for what should or should not be described.

