# OpenReview forum: "On Fairness of Task Arithmetic: The Role of Task Vectors"
_NeurIPS.cc/2025/Workshop/Reliable_ML — NeurIPS 2025 - Reliable ML Workshop_

### Official Review · Reviewer_6Lnx · 2025-09-19
**Review for Fairness of task arithmetic**

**Rating:** 7
**Confidence:** 4

**Review:**

This paper studies a well-motivated and important problem that fits well with the workshop theme: how to adapt LLMs efficiently while also controlling fairness in sensitive tasks like hate-speech detection. The idea of task vectors, parameter differences between base and fine-tuned model, is particularly interesting because it avoids retraining and offers interpretability. Testing fairness metrics in this framework is a very natural research question. The paper is clear and easy to follow, with experiments on LLaMA-2-7B and DistilBERT showing that scaling task vectors with a coefficient $\lambda$ can often maintain accuracy while improving fairness metrics such as DPD and EOD, though effects vary by groups and subgroups. The work’s strengths are its simplicity, the analytical bound linking $\lambda$ to fairness, and the careful comparisons provided to FFT and LoRA. Still, the study appears to fix a uniform $\lambda$ across all subgroups and reports results from relatively small datasets. Fairness metrics could also be presented more precisely. Lastly, the choice of $\lambda$ right now seems somewhat heuristic. Ideally, we would want a more principled way to find a value for $\lambda$ that produces a competitive fairness score while maintaining its accuracy. Overall though, the paper is clear and makes a good contribution.

---

### Official Review · Reviewer_S4rX · 2025-09-20

**Rating:** 6
**Confidence:** 3

**Review:**

**Summary**
The paper studies fairness effects of *task arithmetic* with task vectors, comparing against full fine-tuning (FFT) and LoRA on hate speech detection. Using LLaMA-2 and DistilBERT across gender and race subgroups, it evaluates accuracy, DPD, and EOD, and analyzes scaling coefficients. Results show task arithmetic can sometimes improve fairness without harming accuracy, though effects vary by subgroup. A theoretical bound links scaling to fairness disparity.

**Strengths**
* Solid experimental setup with multiple baselines and metrics.
* Theoretical analysis complements empirical findings.
* Relevant to reliable ML under biased data.

**Weaknesses**
* Motivation gap: The central justification is that fairness under task arithmetic has not been studied. This “research gap” framing feels insufficient without tying to practical reliability concerns (e.g., fairness risks in moderation systems if task arithmetic is deployed).
* Inconsistent findings: Improvements are not universal. Some subgroups (e.g., Men, Asian) benefit from task vector injection, but others (e.g., Women, Native American) see fairness worsen. The paper downplays this instability, yet in practice such subgroup-specific regressions are critical.
* Scope limitations: Experiments are confined to NLP toxicity detection tasks. No evidence is provided for other modalities (vision, speech, multi-modal), leaving generalization unclear.

**Suggestions for Authors**
To resolve the weaknesses.